# Spatial distribution and determinants of solitary childbirth in Ethiopia: Evidence from the 2019 interim demographic and health survey

Tadesse Tarik Tamir[1]*, Berhan Tekeba[1], Alebachew Ferede Zegeye[2], Deresse Abebe Gebrehana[3], Mulugeta Wassie[4], Gebreeyesus Abera Zeleke[5], Enyew Getaneh Mekonen[5]

1 Department of Pediatrics and child Health Nursing, School of Nursing, College of Medicine and Health Sciences, University of Gondar, Gondar, Ethiopia, 2 Department of Medical Nursing, School of Nursing, College of Medicine and Health Sciences, University of Gondar, Gondar, Ethiopia, 3 Department of Internal Medicine, School of Medicine, College of Medicine and Health Sciences, University of Gondar, Gondar, Ethiopia, 4 School of Nursing, College of Medicine and Health Sciences, University of Gondar, Gondar, Ethiopia, 5 Department of Surgical Nursing, School of Nursing, College of Medicine and Health Sciences, University of Gondar, Gondar, Ethiopia

* tadestar140@gmail.com

## Abstract

### Introduction

Solitary childbirth—giving birth without any form of assistance—remains a serious global public health issue, especially in low-resource settings. It is associated with preventable maternal complications such as hemorrhage and sepsis, and poses significant risks to newborns, including birth asphyxia, infection, and early neonatal death. In Ethiopia, where many births occur outside health facilities, understanding the spatial and socio-demographic patterns of solitary childbirth is vital for informing targeted interventions to improve maternal and child health outcomes. This study aims to identify and map the spatial distribution of solitary childbirth across Ethiopia and to analyze its determinants using data from the 2019 national Interim Demographic and Health Survey.

### Method

We analyzed data from the 2019 Interim Ethiopian Demographic and Health Survey to determine the spatial distribution and factors of solitary birth in Ethiopia. A total weighted sample of 3,884 women was included in the analysis. Spatial analysis was used to determine the regional distribution of solitary birth, and multilevel logistic regression was employed to identify its determinants. ArcGIS 10.8 was used for spatial analysis, and Stata 17 was used for multilevel analysis. The fixed effect was analyzed by determining the adjusted odds ratio with a 95% confidence interval.

**Data availability statement:** All relevant data are within the manuscript and its Supporting Information files.

**Funding:** The author(s) received no specific funding for this work.

**Competing interests:** The authors declare that there is no conflict of interest.

**Abbreviations:** AOR, Adjusted Odds Ratio; AIC, Akaike's Information Criterion; BIC, Bayesian Information Criterion; CI, Confidence Interval; IEDHS, Interim Ethiopian Demographic and Health Survey; FDR, False Discovery Rate; ICC: Intra-class correlation coefficient;MOR, Median odds ratio; LL, Log likelihood; LLR, log likelihood ratio; PCV, Proportional Change in Variance; RR, Relative risk; SDHS, Senegal Demographic and Health Survey; WHO, Worled Health Organization.

## Result

The prevalence of solitary childbirths in Ethiopia was 12.73%, with a 95% confidence interval spanning from 11.71% to 13.81%. The western and southern parts of Oromia, all of Benishangul-Gumuz, most parts of the SNNPR, and the west of Amhara regions were hotspot areas for solitary birth. Having no formal education, not attending ANC visits, and residing in pastoral regions were significantly associated with higher odds of solitary birth in Ethiopia.

## Cocnlusion

A notable proportion of women are experiencing childbirth alone, which highlights a significant aspect of maternal health in the country, reflecting both the challenges and improvements in childbirth practices. The distribution of solitary births exhibited spatial clustering with its hotspot areas located in western and southern parts of Oromia, all of Benishangul-Gumuz, most parts of the SNNPR, and west of Amhara regions. Lack of education, not having an ANC visit, and being a resident of pastoral regions were significant determinants of solitary birth. The implementation of maternal and child health strategies in Ethiopia could benefit from considering the hotspot areas and determinants of solitary birth.

## Introduction

Ongoing support during childbirth is linked to better perinatal and infant health outcomes, such as a reduced risk of low birth weight and unplanned cesarean sections, and fewer medical interventions [1,2]. Support persons play a crucial role by communicating with healthcare providers, advocating for the patient's preferences, and offering both physical and emotional support [3]. Solitary childbirth, defined as a woman delivering a baby alone without any form of assistance, is a significant public health concern in Ethiopia [4]. This phenomenon is particularly prevalent in rural and remote areas, where access to healthcare facilities and support systems is limited1 [4].

Globally, maternal mortality remains unacceptably high, with over 260,000 women dying from pregnancy and childbirth-related complications in 2023 alone, and approximately 92% of these deaths occurring in low- and lower-middle-income countries [5]. Unassisted deliveries significantly contribute to this burden, as they are associated with increased risks of postpartum hemorrhage, infections, obstructed labor, and neonatal death [6]. In Ethiopia, the situation is particularly alarming: the maternal mortality rate stood at 412 per 100,000 live births in 2019, with over half of births occurring at home, often without skilled attendance [7]. These outcomes underscore the need to understand the geographic distribution and determinants of solitary childbirth to inform targeted interventions and reduce preventable maternal and child deaths.

Solitary childbirth remains a significant barrier to achieving global maternal and neonatal health targets, particularly those outlined under Sustainable Development Goal 3 (SDG 3) [8]. SDG 3 aims to "ensure healthy lives and promote well-being for

all at all ages," with specific objectives to reduce the global maternal mortality ratio to less than 70 per 100,000 live births and to end preventable deaths of newborns and children under five years of age by 2030 [8,9]. These targets include reducing neonatal mortality to at least 12 per 1,000 live births and under-five mortality to at least 25 per 1,000 live births [8,9].

Several factors contribute to the high incidence of solitary childbirth in Ethiopia. These include socio-economic determinants such as education level, media exposure, household income, and cultural practices [10–12]. Women with lower educational attainment and those from poorer households are more likely to give birth alone [1]. Additionally, cultural norms and beliefs about childbirth, including the preference for traditional birth practices and the stigma associated with hospital births, play a significant role [10–12].

Ethiopia is characterized by diverse geographical and socio-economic conditions that influence health service accessibility. The country's healthcare system has made strides in improving maternal health services; however, disparities remain, particularly in rural regions [13–15]. Geographical barriers also significantly impact the spatial distribution of solitary status of childbirth. Regions with difficult terrain and limited infrastructure, such as the Afar and Somali regions, exhibit higher rates of home births without skilled attendants [3,15]. The distance to healthcare facilities and the availability of transportation are critical factors influencing a woman's decision to seek professional care during childbirth. Moreover, the availability and quality of maternal health services vary across different regions of Ethiopia. Urban areas, such as Addis Ababa, generally have better access to healthcare facilities and skilled birth attendants compared to rural areas [4]. This urban-rural divide highlights the need for region-specific strategies to address the unique challenges faced by women in different parts of the country.

To the best of our knowledge, this study is the first of its kind in Ethiopia to specifically examine the phenomenon of solitary childbirth—defined as giving birth without any form of assistance. While previous research has primarily focused on the type of birth assistance (skilled versus traditional) or the place of delivery (home versus health facility), the unique and critical issue of women delivering entirely alone has received little to no scholarly attention. This gap is particularly concerning given that solitary childbirth remains a common practice in various parts of Ethiopia, often rooted in cultural traditions that encourage women to give birth in isolation or driven by the unavailability of assistance during labor. Addressing this overlooked aspect of maternal health is essential for informing targeted interventions and improving outcomes for both mothers and newborns. Therefore, analyzing the spatial distribution and associated factors of solitary childbirth in Ethiopia is essential for identifying high-risk areas and populations. This analysis can inform the development of targeted interventions aimed at reducing maternal and neonatal mortality and improving overall maternal health outcomes. By addressing the socio-economic, cultural, and geographical barriers to skilled birth attendance, Ethiopia can make significant progress towards achieving its maternal health goals.

## Methods and materials

### Study design, data source and setting

A secondary data analysis of cross-sectional 2019 Interim Ethiopian Demographic and Health Survey (IEDHS) was conducted. The IEDHS data were accessed through the Monitoring and Evaluation to Assess and Use Results Demographic and Health Survey (MEASURE DHS) program. Specifically, we utilized the Individual Recode (IR) data extracted from the IEDHS dataset. This study was conducted in Ethiopia, located in the Horn of Africa. The country is administratively structured into nine regional states—Tigray, Afar, Amhara, Oromia, Somali, Benishangul-Gumuz, Southern Nations, Nationalities, and Peoples' Region (SNNPR), Gambela, and Harari—and two chartered city administrations: Addis Ababa and Dire Dawa [16]. Ethiopia operates a decentralized health system organized into three tiers [17,18]. The primary level includes health posts, health centers, and primary hospitals, which serve as the first point of contact for communities and provide essential health services. A primary hospital typically serves a population of around 100,000 and functions as a referral

center for health centers, offering emergency, outpatient, and inpatient care. The secondary level comprises general hospitals that act as referral hubs for primary hospitals and serve as training institutions for healthcare professionals. The tertiary level consists of specialized hospitals that provide advanced care and serve as referral centers for general hospitals This structured health system is designed to improve access to care across both urban and rural populations [17,18].

**Population and sampling procedure.** The source population for this study consisted of women aged 15–49 years in Ethiopia who gave birth three years preceding the survey. The study population, on the other hand, comprised women aged 15–49 years who had given birth within the three years preceding the survey and were residing in the enumeration areas covered by the survey. The IEDHS employed a stratified two-stage cluster sampling technique to ensure nationally representative data. Firstly, each country is divided into different strata based on relevant characteristics, such as urban/rural location or geographic regions. Within each stratum, enumeration areas (clusters) were randomly selected as the primary sampling units. In the second stage, a sample of households from within each selected enumeration area using either a systematic method was drawn [7]. Sampling weights were applied using the svyset and svy commands to account for the complex survey design of the IEDHS data. This involved utilizing weighting variables, including the sampling weight (v005), primary sampling unit (v021), and strata for sampling design (v023). This approach ensures that the sample is representative of the target population. By incorporating these sampling weights, we aimed to adjust for the unequal probabilities of selection inherent in the survey design, thereby enhancing the accuracy and validity of our findings and allowing for the drawing of valid conclusions. In this study, a total weighted sample of 3884 women were included (Fig 1). ArcGIS 10.8 was used for spatial analysis, and Stata 17 was used for multilevel analysis.

## Variables and measurement

In this study, the outcome variable was defined as solitary childbirth. Solitary childbirth refers to the event where a woman gives birth without any assistance from healthcare professionals, midwives, family members, or any other individuals. This includes births that occur in any setting where the mother was entirely alone during the delivery process. The variable was measured as a binary outcome, with a value of 1 indicating a solitary childbirth (no assistance present) and a value of 0

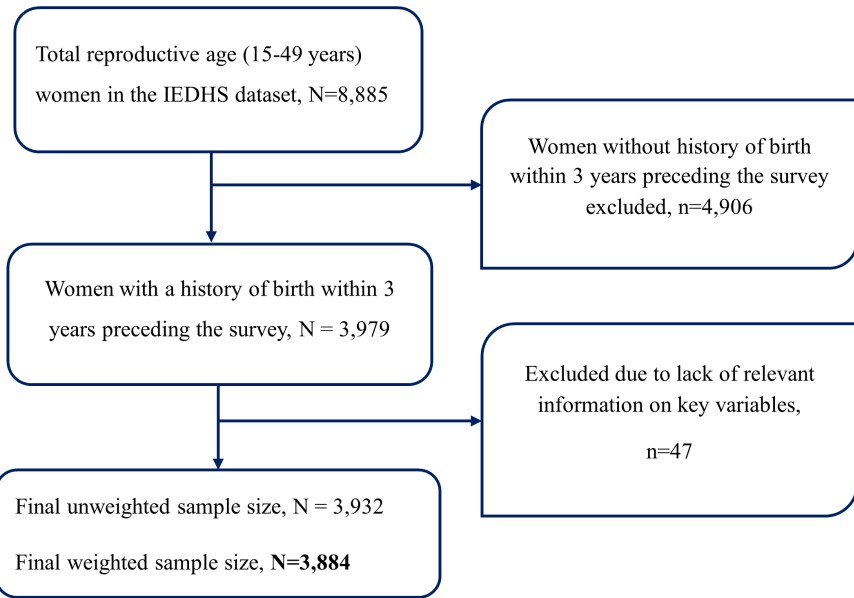

**Fig 1. Schematic depiction of the sample size determination of the study.**

indicating a non-solitary childbirth (assistance present). According to the Guide to DHS Statistics, the types of persons providing delivery assistance are categorized as follows: doctor (m3a), nurse/midwife (m3b), auxiliary midwife (m3c), traditional birth attendant (m3g), relative/other (m3h, m3i, m3j, m3k, m3l, m3m), and no one (m3n) [19]. For this study, solitary childbirth corresponds to the category where no one (m3n) is present. The data for this variable is sourced from the 2019 IEDHS, which includes self-reported information on the presence or absence of assistance during childbirth.

The explanatory variables for this study were chosen based on established guidelines [20] and scholarly literature. These variables were divided into two levels. At the individual level, we included factors such as age, educational attainment, marital status, antenatal care utilization, birth interval, presence of a radio, presence of a TV, sex of the household head, household size, and wealth index. At the community level, we analyzed variables such as residence type, region, women's illiteracy rate, community poverty rate, and regional characteristics.

**Wealth index in the 2019 IEDHS.** Households are classified according to the number and types of consumer goods they possess, which can include items like televisions, bicycles, and automobiles, as well as housing characteristics such as drinking water sources, bathroom facilities, and flooring materials. Principal component analysis is employed to derive these scores. National wealth quintiles are determined by assigning the household score to each usual household member, ranking individuals based on their scores, and dividing the population into five equal segments (poorer, poorest, middle, richer, and rich), each representing 20% of the population [7].

## Spatial analysis

**Spatial autocorrelation.** The spatial dependency of solitary birth in Ethiopia was assessed using Global Moran's I, a measure of spatial autocorrelation. This statistic ranges from −1–1, where a value of 0 indicates random distribution, a value near −1 suggests a dispersed pattern, and a value close to 1 signifies clustering. If the Moran's I value is statistically significant ($P < 0.05$), it indicates the presence of spatial dependence [21].

**Hot spot analysis.** This study employed an optimized hot spot analysis technique to pinpoint areas with high and low solitary birth rates in Ethiopia. The Optimized Hot Spot Analysis tool, which uses the Getis-Ord Gi* statistic, is an advanced version of the traditional Hot Spot Analysis (Getis-Ord Gi*). It identifies statistically significant hot and cold spots in the data and adjusts for multiple testing and spatial dependence using the False Discovery Rate (FDR) correction method [22]. The Getis-Ord Gi* statistic measures spatial clustering by analyzing the distribution of features and their neighboring features [22,23].

**Spatial scan statistical analysis.** To identify significant clusters of solitary birth, Scan Statistical Analysis were utilized, employing a circular scanning window that moves across the study area. The analysis incorporated cases, controls, and geographic coordinate data into the Bernoulli model. For each potential cluster, the Log Likelihood Ratio (LLR), Relative Risk (RR), and P-values were calculated to determine if the observed number of cases within the cluster was significantly higher than expected.

**Spatial interpolation.** The spatial prediction of solitary birth in unsampled areas of Ethiopia was conducted using Kriging interpolation, based on observed data from sampled regions. Kriging is an interpolation method that estimates the value of a variable in unsampled locations by utilizing observations from neighboring areas [24–26]. This technique minimizes prediction errors and represents geographic variation through a variogram. Named after Danie Krige, Kriging originated in the field of mining geology [25,26].

**Multilevel modeling.** This study utilized data from the Demographic and Health Surveys (DHS), which provide information at both household and cluster levels (hierarchical in nature). To tackle the problem of non-independent observations—an essential requirement for standard logistic regression—we employed mixed-effects models with a binary outcome variable. We developed four distinct model specifications: a null model to evaluate random effects and confirm the suitability of multilevel regression; Model I, which included the outcome variable and first-level control variables; Model II, which added second-level control variables; and a comprehensive Model III that encompassed all variables—outcome,

first-level controls, and second-level controls. This multilevel modeling approach enabled us to account for the hierarchical structure of the DHS data and more effectively investigate the determinants of solitary childbirth. The regression model was equated as follows [27]:

$$\log\left(\frac{\Pi ij}{1-\Pi ij}\right) = \beta 0 + \beta 1 x 1 ij + ... + \beta n x n ij + \gamma 0 + \gamma 1 z 1 ij + ... + \gamma m z m ij + u o ij$$

Where πij represents the probability of solitary birth for the $i^{th}$ women in the $j^{th}$ cluster, while (1-πij) denotes the probability of the $i^{th}$ women in the $j^{th}$ cluster not experiencing solitary birth. The intercept term β0 characterizes the baseline of the regression equation. The coefficients β1 to βn are linked to the level 1 variables x1ij to xnij, which exert an influence on the response variable at the individual level. The intercept γ0j captures the random effect at level 2, while the coefficients γ1 to γm are associated with the level 2 variables z1ij to zmij, reflecting cluster-level effects. Finally, the error term eij accounts for the random error or residual within the model.

The researchers evaluated both the fixed and random components of the mixed-effects models. The random effect was assessed using variance, the intra-class correlation coefficient (ICC), the median odds ratio (MOR), and the proportional change in variation (PCV). The fixed effect was analyzed by determining the adjusted odds ratio (AOR) with a 95% confidence interval (CI). An association between explanatory variables and solitary childbirth was considered significant if the p-value was below the predetermined significance level of 0.05. For model comparison, the team employed log likelihood, deviance, Akaike's Information Criterion (AIC), and Bayesian Information Criterion (BIC).

To address multicollinearity—when two or more independent variables in a regression model are highly correlated—the researchers calculated the variance inflation factor (VIF) for each variable, finding VIF values below five, indicating that multicollinearity was not a significant issue. Additionally, multivariable regression techniques were utilized to control for potential confounding factors, allowing researchers to isolate the effect of each independent variable on solitary childbirth while accounting for other relevant influences.

**Ethical approval.** This study was based on analysis of existing survey dataset in the public domain that are freely available online with all the identifier information anonymized, no ethical approval was required. The first author obtained authorization for the download and usage of the archive of the IEDHS dataset from MEASURE DHS. The datasets were treated with the utmost confidentiality, and issues related to informed consent, anonymity, and privacy was ethically handled by the MEASURE DHS office. We did not manipulate or apply the microdata beyond the scope of this study.

## Results

### Descriptive statistics of solitary births by individual and community-level characteristics

The analysis included a total weighted sample of 3,884 women within three years preceding the survey. Table 1 summarizes the association between individual and community-level variables and the occurrence of solitary births. Among individual-level factors, the proportion of solitary births is highest among women aged 35–49 years (17.59%), while the lowest is observed in the 15–19 age group (2.04%), with this association being statistically significant ($p < 0.001$). Marital status does not show a significant difference in solitary births, with 12.92% of women not in union and 12.71% of those in union reporting solitary births ($p = 0.430$). Educational level, however, is significantly associated with solitary births; women with no education exhibit the highest proportion (18.16%), compared to only 1.13% among those with secondary or higher education ($p < 0.001$). Similarly, women in male-headed households report higher rates of solitary births (12.98%) compared to those in female-headed households (10.99%) ($p < 0.001$). Antenatal care (ANC) visits also play a critical role, with women who had no ANC visits reporting a markedly higher rate of solitary births (22.23%) than those who received ANC (9.42%) ($p < 0.001$). Household wealth is inversely associated with solitary births; the poorest households exhibit

the highest rates (20.99%), while the richest households show the lowest (3.12%) (p < 0.001). Additional individual-level factors significantly associated with solitary births include household size, access to media (radio and television), and birth interval.

**Table 1. Descriptive statistics of solitary births by individual and community-level characteristics.**

| Characteristics | | Solitary birth | | Total | Chi squared test P value |
|---|---|---|---|---|---|
| | | Yes [n (%)] | No [n (%)] | | |
| **Individual level characteristics** | | | | | |
| Women age | 15-19 | 5 (2.04) | 222 (97.96) | 227 (5.82) | <0.001 |
| | 20-34 | 325 (11.95) | 2399 (88.05) | 2724 (70.16) | |
| | 35-49 | 164 (17.59) | 769 (82.41) | 933 (24.02) | |
| Marital status | Not in union | 30 (12.92) | 202 (87.08) | 232 (5.99) | 0.430 |
| | In union | 464 (12.71) | 3187 (87.29) | 3651 (94.01) | |
| Women educational level | No education | 364 (18.16) | 1639 (81.84) | 2003 (51.57) | <0.001 |
| | Primary | 125 (8.94) | 1274 (91.06) | 1,399 (36.03) | |
| | Secondary & above | 5 (1.13) | 476 (98.87) | 482 (12.40) | |
| Sex of household head | Male | 439 (12.98) | 2940 (87.02) | 3379 (87.01) | <0.001 |
| | Female | 55 (10.99) | 449 (89.01) | 505 (12.99) | |
| ANC visits | No visit | 223 (22.23) | 779 (77.77) | 1002 (25.80) | <0.001 |
| | Had visit | 272 (9.42) | 2610 (90.58) | 2882 (74.20) | |
| Household wealth index | Poorest | 173(20.99) | 651 (79.01) | 824 (21.22) | <0.001 |
| | Poorer | 134 (16.48) | 680 (83.52) | 814 (20.96) | |
| | Middle | 84 (11.24) | 667 (88.76) | 752 (19.36) | |
| | Richer | 77 (11.23) | 612 (88.77) | 689 (17.75) | |
| | Richest | 25 (3.12) | 779 (96.88) | 804 (20.71) | |
| Birth interval | < 24 months | 137 (19.05) | 581 (80.95) | 717 (30.50) | 0.116 |
| | ≥ 24 months | 281 (17.18) | 1354 (82.82) | 1635 (69.50) | |
| Household has radio | Yes | 85 (8.26) | 949(91.74 | 1035 (26.64) | <0.001 |
| | No | 409 (14.35) | 2440 (85.65) | 2849 (73.36) | |
| Household has television | Yes | 12 (1.82) | 630 (98.18) | 642 (16.52) | <0.001 |
| | No | 483 (14.88) | 2760 (85.12) | 3242 (83.48) | |
| Household size | < 6 | 255 (9.93) | 2312 (90.07) | 2567 (66.10) | <0.001 |
| | ≥6 | 239 (18.18) | 1077 (81.82) | 1317 (33.90) | |
| **Community level characteristics** | | | | | |
| Residence | Urban | 60 (5.91) | 956 (94.09) | 1016 (26.16) | <0.001 |
| | Rural | 434 (15.14) | 2434 (84.86) | 2868 (73.84) | |
| Community illiteracy rate | Low | 204 (9.90) | 1860(90.10) | 2064 (53.14) | <0.001 |
| | High | 290 (15.93) | 1530 (84.07) | 1820 (46.86) | |
| Community poverty rate | Low | 234 (9.76) | 2161 (90.24) | 2395 (61.66) | <0.001 |
| | High | 260 (17.49) | 1229 (82.51) | 1489 (38.34) | |
| Region | Metropolitan | 1 (0.90) | 155 (99.10) | 157 (4.03) | <0.001 |
| | Agrarian | 477 (14.05) | 2916 (85.95) | 3393 (87.37) | |
| | Pastorals | 50 (14.97) | 284 (85.03) | 334 (8.60) | |

ANC: Antenatal Care.

At the community level, solitary births are more prevalent in rural areas (15.14%) compared to urban areas (5.91%) (p<0.001). Communities with higher illiteracy rates report a higher proportion of solitary births (15.93%) compared to those with lower illiteracy rates (9.90%), and similar patterns are observed in relation to community poverty rates, where communities with higher poverty rates have a significantly higher proportion of solitary births (17.49%) compared to their counterparts with lower poverty rates (9.76%) (p<0.001). Regional disparities are evident, with the highest proportion of solitary births observed in pastoral settings (14.97%) and the lowest in metropolitan areas (0.90%) (p<0.001) (Table 1).

These findings indicate significant disparities in solitary birth rates based on socioeconomic, demographic, and geographic factors. Most of these associations are statistically significant (p<0.05), underscoring the need for targeted interventions that address these determinants.

### Prevalence of solitary childbirth in Ethiopia

The prevalence of solitary childbirths in Ethiopia was 12.73%, with a 95% confidence interval spanning from 11.71% to 13.81%, as illustrated in Fig 2.

**Spatial autocorrelation of solitary birth in Ethiopia.** The spatial autocorrelation analysis indicated significant variation in the distribution of solitary births across Ethiopia, with a Moran's Index of 0.169016, a Z-score of 3.776489, and a p-value of 0.000159 (Fig 3). This suggests that solitary births are not uniformly distributed throughout the country.

**Hot spot analysis of solitary birth in Ethiopia.** Fig 4 illustrates the results of the optimized hot spot analysis of solitary births in Ethiopia. This analysis identifies statistically significant clusters of high and low values (hot spots and cold spots) across the country. The map is color-coded: red indicates hot spots with 99% confidence, orange represents 95% confidence, and blue signifies cold spots at the same confidence levels. Gray spots denote regions where the analysis did not yield significant results. The findings highlight specific regions, such as the western and southern parts of Oromia, throughout Benishangul-Gumuz, most parts of SNNPR, and the northwest of Amhara, where solitary births are concentrated (hot spot areas).

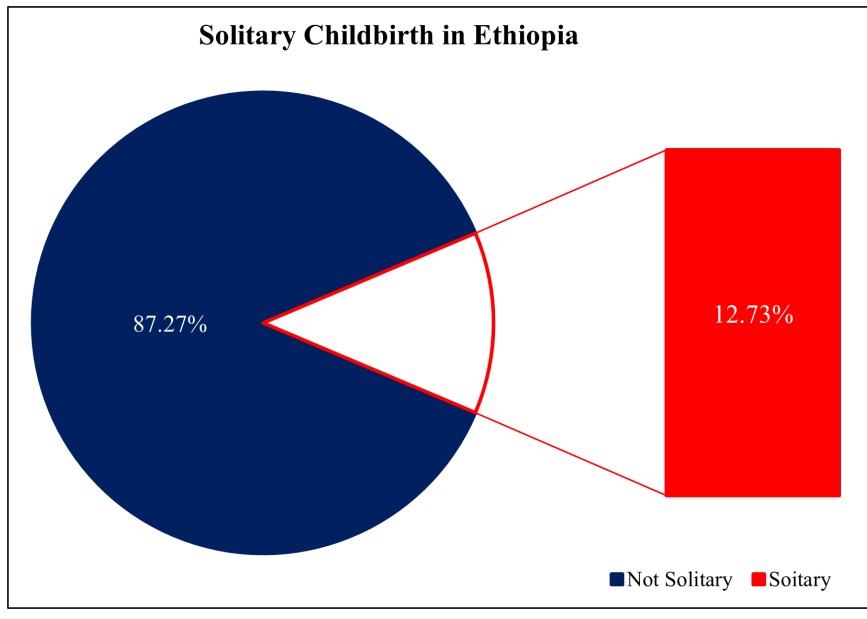

**Fig 2. Prevalence of solitary childbirths in Ethiopia, IEDHS 2019.**

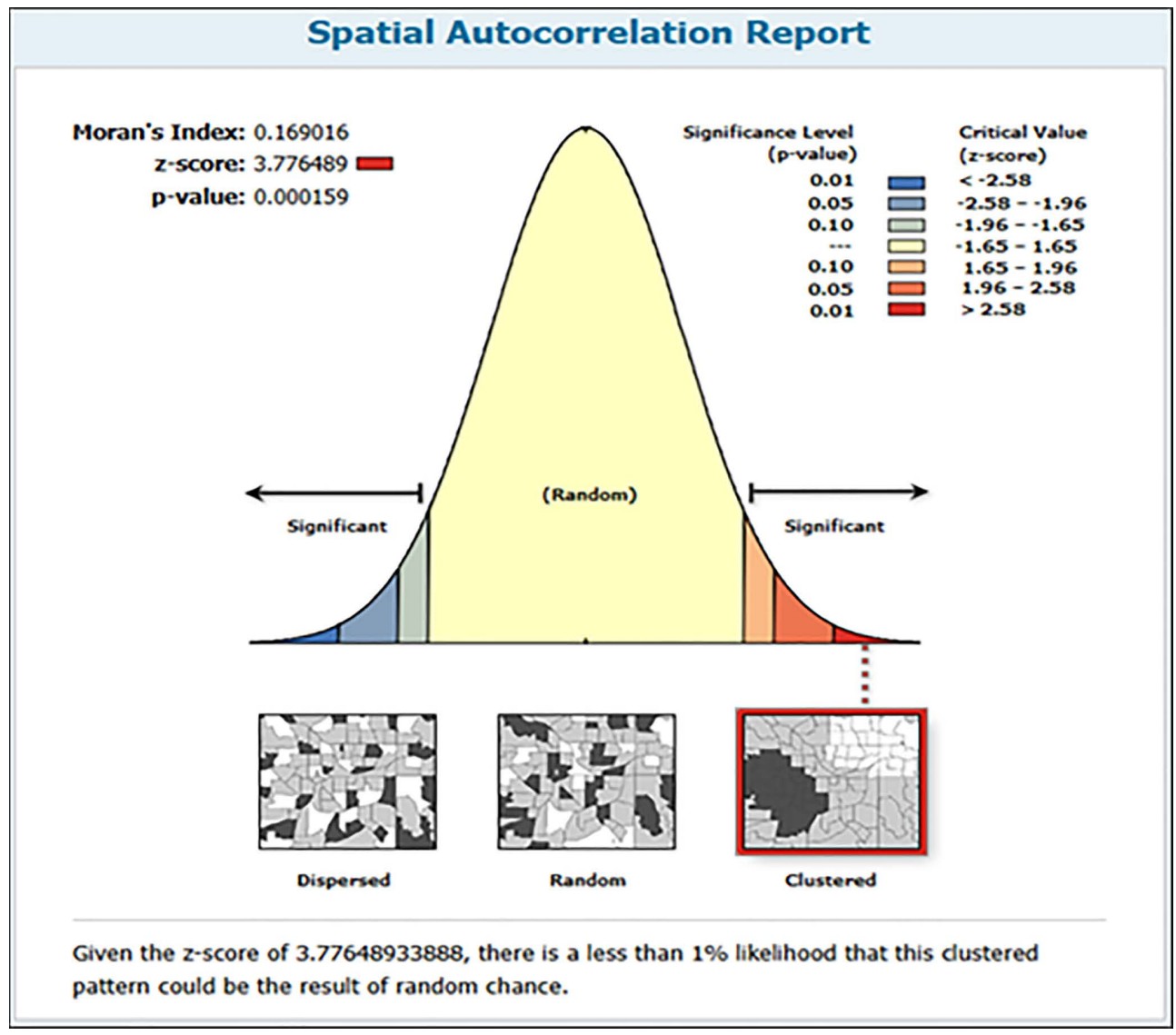

**Fig 3. Spatial autocorrelation of solitary birth in Ethiopia, IEDHS 2019.**

**Spatial scan statistical analysis of solitary birth in Ethiopia.** In this study, we conducted a spatial scan statistical analysis of solitary births in Ethiopia, as illustrated in Table 2 and Fig 5. The analysis identified several clusters of solitary births, with the primary cluster comprising 45 enumeration areas located at a latitude of 5.0000° N and a longitude of 37.0000° E, covering a radius of 333.38 km. This primary cluster had a total population of 1008, with 548 recorded cases of solitary births, resulting in a relative risk (RR) of 2.68, indicating a 2.68 times higher risk of solitary birth compared to the women living outside the cluster window. The LLR for this cluster was 67.27, suggesting strong evidence of spatial clustering, and the associated p-value was less than 0.001, confirming the statistical significance of this finding. Additionally, three secondary clusters were identified, with Secondary Cluster 1 encompassing 17 enumeration areas and exhibiting LLR of 46.48. Secondary Cluster 2 included 21 areas with an RR of 2.37, while Secondary Cluster 3 comprised

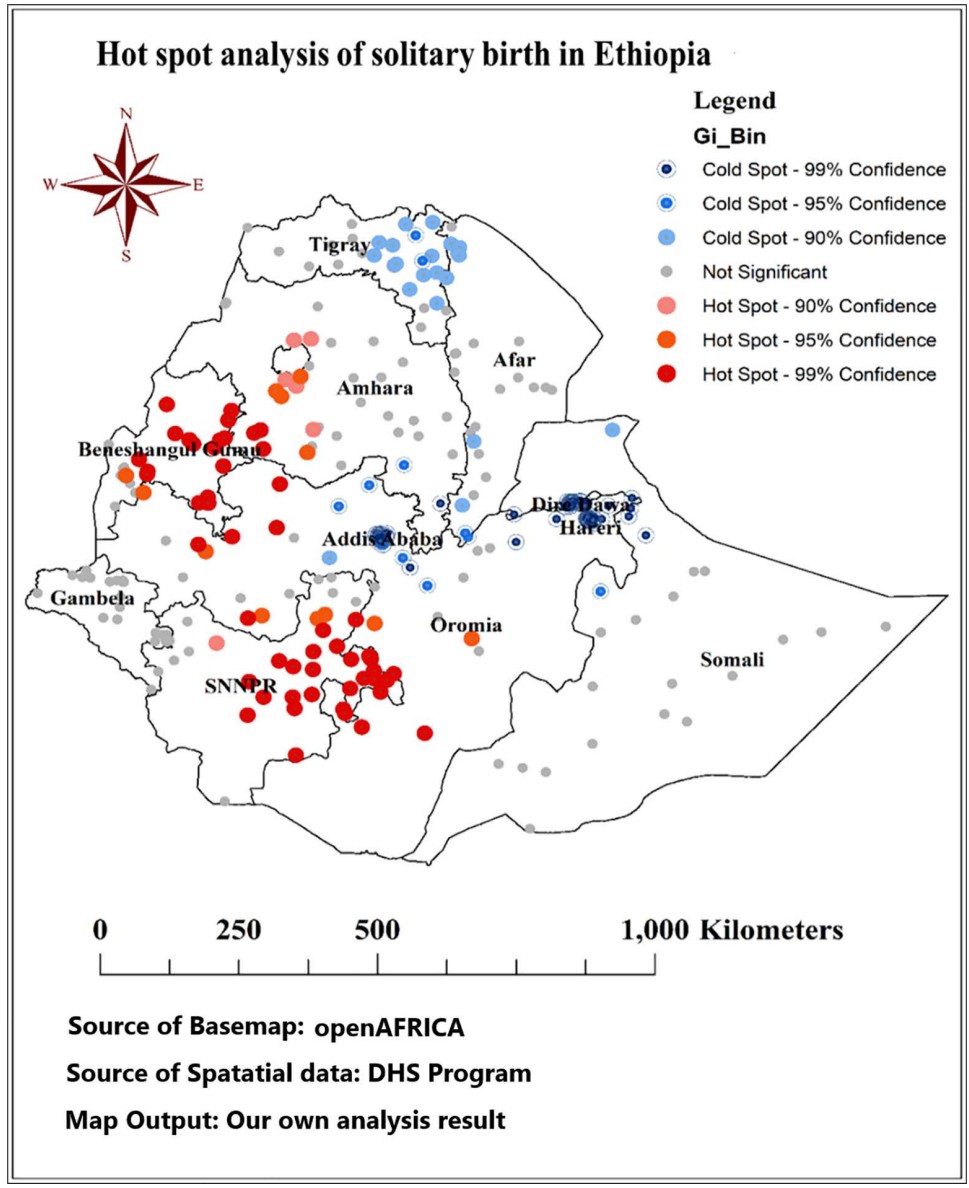

**Fig 4. Hot spot analysis of solitary birth in Ethiopia, IEDHS 2019; Source of basemaps URL:**https://open.africa/dataset/ethiopia-shapefiles**; the figure is similar but not identical to the original image and is therefore for illustrative purposes only.**

9 areas with a lower risk. The spatial distribution of these clusters is depicted in Fig 5, which highlights the geographic context, particularly around key regions such as Oromia and SNNPR.

**Interpolation of solitary birth in Ethiopia.** The Kriging interpolation of solitary births across Ethiopia provides a visual representation of predicted risk levels associated with solitary births in various regions. The color gradient ranges from low (indicated in green) to high (represented in red) risk areas, with specific attention to Benishangul-Gumuz, Oromia, SNNPR, and the eastern part of Afar. These regions exhibit varying degrees of risk. The interpolation method employed offers an estimate of solitary birth occurrences, leveraging spatial data to fill in gaps between observed data points. The

**Table 2. Spatial scan statistical analysis of solitary birth in Ethiopia, IEDHS 2019.**

| Cluster | N | Latitude | Longitude | Radius | Population | Cases | RR | LLR | p-value |
|---|---|---|---|---|---|---|---|---|---|
| Primary | 45 | 5.000000 N | 37.000000 E | 333.38 km | 1008 | 548 | 2.68 | 67.27 | <0.001 |
| Secondary 1 | 17 | 5.000000 N | 37.000000E | 247.57 km | 573 | 150 | 2.53 | 46.48 | <0.001 |
| Secondary 2 | 3 | 8.000000 N | 37.000000 E | 0 km | 118 | 34 | 2.37 | 11.24 | <0.001 |
| Secondary 3 | 14 | 7.000000 N | 40.000000 E | 156.48 km | 477 | 90 | 1.60 | 8.58 | 0.007 |

N: Number of enumeration areas; RR: Relative risk; LLR: Log likelihood ratio.

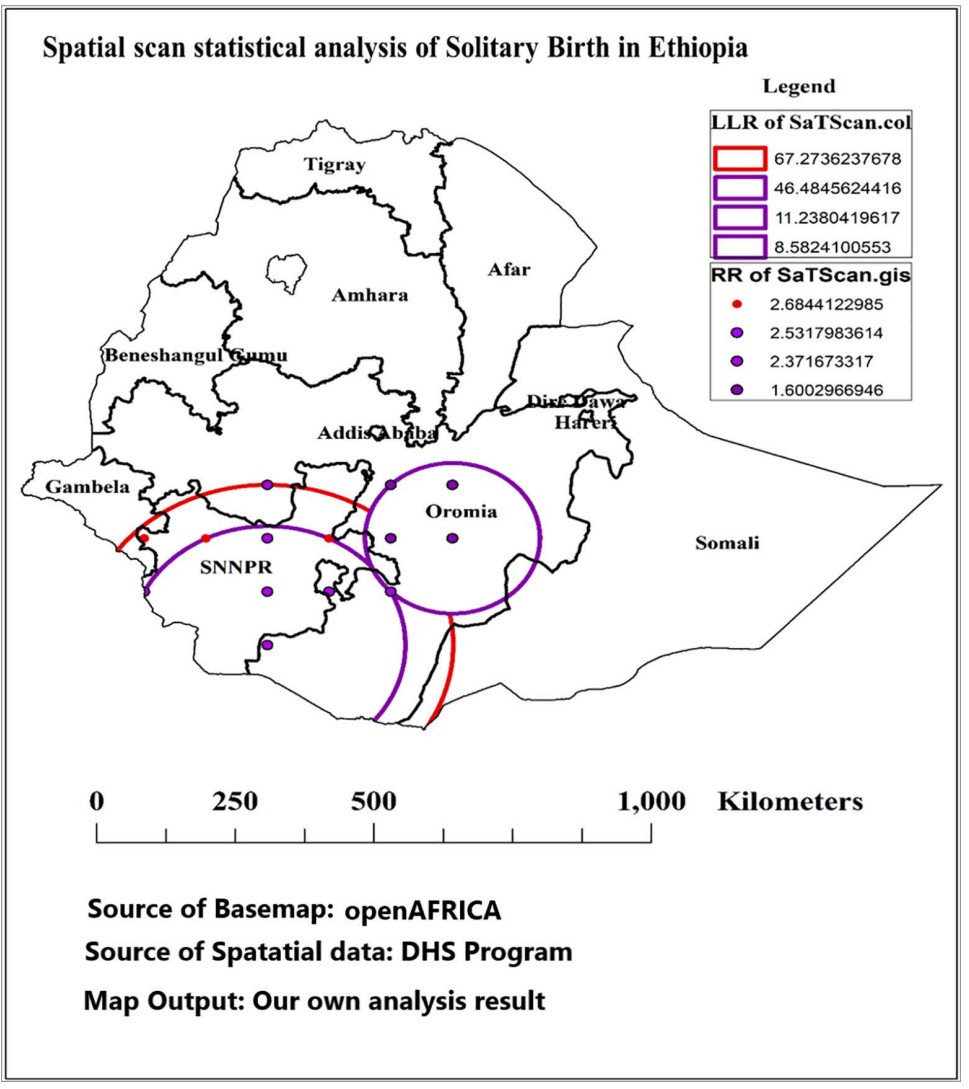

**Fig 5. Spatial scan statistical analysis of solitary birth in Ethiopia, IEDHS 2019; Source of basemaps URL:** https://open.africa/dataset/ethiopia-shapefiles**; the figure is similar but not identical to the original image and is therefore for illustrative purposes only.**

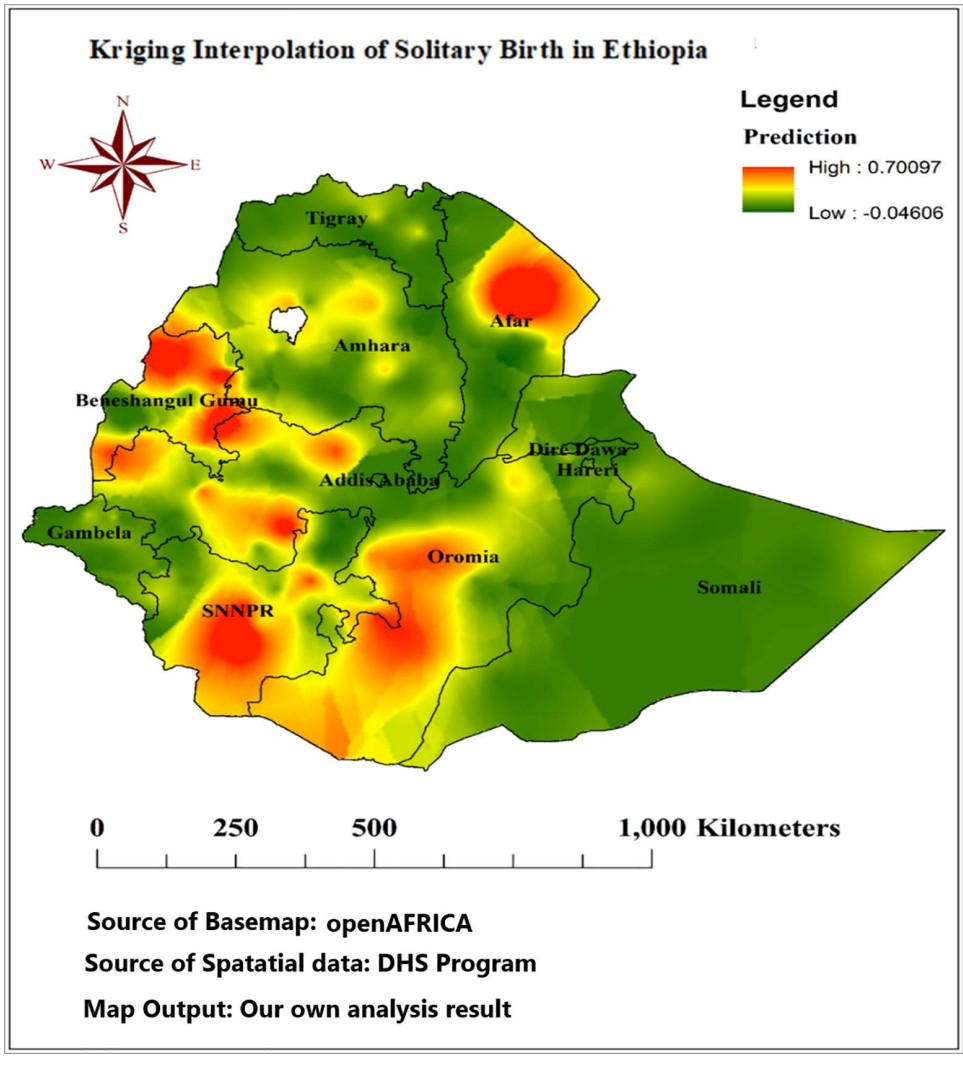

**Fig 6. Kriging interpolation of solitary birth in Ethiopia, EIDHS 2019; Source of basemaps URL:** https://open.africa/dataset/ethiopia-shapefiles; the figure is similar but not identical to the original image and is therefore for illustrative purposes only.

legend clearly delineates the predicted risk levels, with a higher prediction value of 0.70097 indicating areas of greater concern, while a lower value of −0.04606 suggests minimal risk (Fig 6).

**Multivariable multilevel logistic regression of determinants of solitary birth in Ethiopia.** This study employed various metrics to assess model fitness. The log likelihood values indicate a progression from the Null model (−1002.385) to Model III (−690.79), suggesting an improved fit with increasingly complex models. Deviance values also decreased, from 2,004.77 in the Null model to 1,381.58 in Model III, further supporting this trend. Additionally, the Akaike Information Criterion (AIC) and Bayesian Information Criterion (BIC) show reductions as model complexity increases, with AIC values decreasing from 2008.77 for the Null model to 1423.57 for Model III, and BIC values dropping from 2021.32 to 1530.66. These findings indicate that more complex models provide a better fit to the data while accounting for the number of parameters. In this regard, Model III was determined to be the best fit for our data (Table 3).

**Table 3. Multivariable multilevel logistic regression of determinants of solitary birth in Ethiopia.**

**Model fitness**

| Metrics | Null model | Model I | Model II | Model III |
|---|---|---|---|---|
| Log likelihood | −1002.385 | −703.21 | −978.03 | −690.79 |
| Deviance | 2,004.77 | 1,406.42 | 1,956.06 | 1,381.58 |
| AIC | 2008.77 | 1438.422 | 1970.07 | 1423.57 |
| BIC | 2021.32 | 1544.64 | 2014.01 | 1530.66 |

**Measures of Association (Fixed effect)**

| Factors | | Model I | Model II | Model III |
|---|---|---|---|---|
| Women age | 15-19 | 2.89 (0.42, 20.09) | | 3.16 (0.47, 21.31) |
| | 20-34 | 1.00 | | 1.00 |
| | 35-49 | 1.02 (0.73, 1.43) | | 0.96 (0.69, 1.35) |
| Marital status | Not in union | 1.00 | | 1.00 |
| | In union | 0.70 (0.29, 1.65) | | 0.75 (0.31, 1.80) |
| Women educational level | No education | 4.67 (1.23, 17.76) | | **4.43 (1.15, 17.05)\*** |
| | Primary | 2.64 (0.69, 10.07) | | 2.41 (0.63, 9.25) |
| | Secondary & above | 1.00 | | 1.00 |
| Sex of household head | Male | 1.56 (0.87, 2.80) | | 1.34 (0.74, 2.43) |
| | Female | 1.00 | | 1.00 |
| ANC visits | No visit | 2.27 (1.60, 3.20) | | **2.42 (1.71, 3.43)\*** |
| | Had visit | 1.00 | | 1.00 |
| Household wealth index | Poorest | 2.53 (0.80, 7.95) | | 2.64 (0.79, 8.84) |
| | Poorer | 2.15 (0.69, 6.78) | | 1.93 (0.58, 6.41) |
| | Middle | 1.89 (0.60, 5.92) | | 1.55 (0.48, 5.06) |
| | Richer | 1.42 (0.46, 4.37) | | 1.19 (0.38, 3.74) |
| | Richest | 1.00 | | 1.00 |
| Birth interval | < 24 months | 1.20 (0.86, 1.68) | | 1.27 (0.91, 1.77) |
| | ≥ 24 months | 1.00 | | 1.00 |
| Household has radio | Yes | 1.00 | | 1.00 |
| | No | 1.17 (0.76, 1.79) | | 1.16 (0.75, 1.78) |
| Household has television | Yes | 1.00 | | 1.00 |
| | No | 2.47 (0.70, 8.63) | | 1.94 (0.56, 6.80) |
| Household size | < 6 | 1.00 | | 1.00 |
| | ≥6 | 0.95(0.68, 1.32) | | 0.96 (0.69, 1.34) |
| Residence | Urban | | 1.00 | 1.00 |
| | Rural | | 3.51 (1.37, 9.00) | 2.10 (0.68, 6.52) |
| Community illiteracy rate | Low | | 1.00 | 1.00 |
| | High | | 1.39 (0.75, 2.55) | 0.68 (0.35, 1.34) |
| Community poverty rate | Low | | 1.00 | 1.00 |
| | High | | 1.40 (0.73, 2.67) | 0.67 (0.32, 1.39) |
| Region | Metropolitan | | 1.00 | 1.00 |
| | Agrarian | | 1.55 (0.58, 4.18) | 1.06 (0.34, 3.29) |
| | Pastorals | | 4.67 (1.83, 11.94) | **3.77 (1.28, 11.17)\*** |

**Measures of Variation (Random effects)**

| Parameters | Null model | Model I | Model II | Model III |
|---|---|---|---|---|
| Variance (95% CI) | 3.862 (2.657, 5.614) | 3.507 (2.340, 5.266) | 3.120 (2.078, 4.678) | 3.013 (2.071, 4.384) |
| ICC (%, 95% CI) | 54.00 (44.68, 6305) | 51.63 (41.59, 63.05) | 48.66 (38.71-58.71) | 47.805 (38.63, 57.13) |
| MOR (95% CI) | 6.47 (4.70, 9.50) | 5.92 (4.28, 8.90) | 5.35 (3.93, 2.05) | 5.20 (3.92, 7.31) |

*(Continued)*

**Table 3.** (Continued)

| Model fitness | | | | |
|---|---|---|---|---|
| Metrics | Null model | Model I | Model II | Model III |
| PCV (%) | Reference | 9.19 | 19.21 | 21.98 |

ANC: antenatal care, AOR: Adusted Odds Ratio, AIC: Akaike's Information Criterion, BIC: Bayesian Infromation Criterion, CI: Confidence Interval, ICC: Intra-Class Correlation, MOR: Median Odds Ratio, PCV: Proportional Change in Variation.

The results from the fixed effects section of the mixed-effects model revealed that having no formal education, not attending antenatal care (ANC) visits, and residing in pastoral regions were significantly associated with higher odds of solitary birth in Ethiopia. Specifically, the odds of solitary birth were 4.43 times higher among women with no formal education (Adjusted Odds Ratio [AOR] = 4.43, 95% CI: 1.15, 17.05) compared to those with secondary or higher education. Additionally, women who did not attend any ANC visits during their pregnancy had 2.42 times higher odds of solitary birth (AOR = 2.42, 95% CI: 1.71, 3.43) compared to those who did. Furthermore, the likelihood of solitary birth was 3.77 times higher among women living in pastoral regions of Ethiopia (AOR = 3.77, 95% CI: 1.28, 11.17) compared to those residing in metropolitan regions (Table 3).

The random effects section provides a detailed examination of the measures of variation related to solitary childbirth in Ethiopia, highlighting changes in parameters across different statistical models. The Null model indicates a variance of 3.362 (95% CI: 2.257, 5.614), establishing a baseline for understanding variability in solitary childbirth rates across clusters. The intraclass correlation coefficient (ICC) for the Null model is reported at 54.00% (95% CI: 44.18, 63.51), reflecting that a substantial portion of the variability can be attributed to differences between clusters. This percentage evolves across models, illustrating how the addition of predictors influences the clustering effect.

As the models progress from the Null model to Model III, significant changes in unexplained heterogeneity were observed. The median odds ratio (MOR) decreased from 6.47 in the Null model to 5.20 in Model III. This reduction indicates that the variability in solitary childbirth outcomes attributable to cluster-level factors diminishes as more predictors are incorporated. Such a decrease suggests that additional covariates in Model III more effectively explain the differences in solitary childbirth rates, accounting for contextual factors such as socioeconomic status, access to healthcare, and cultural influences. The proportional change in variance (PCV) helps quantify the extent to which each model improves the explanation of variance in solitary childbirth, reinforcing the importance of incorporating relevant contextual factors (Table 3).

**Potential biases in the study and how they were Addressed.** Several types of bias could potentially affect this study. First, the data on childbirth experiences were self-reported by women and may be subject to inaccuracies due to social desirability. To minimize this, we relied on standardized data collection procedures used in the 2019 IEDHS, which are designed to enhance reliability and reduce reporting errors. Second, misclassification bias could occur if solitary childbirth is inaccurately reported or misunderstood by respondents. To mitigate this, we used a clear operational definition based on DHS coding, distinguishing solitary childbirth from other forms of delivery assistance. Moreover, confounding is a possibility in observational studies like ours. To address this, we employed multivariable multilevel modeling, which allows for the adjustment of both individual- and community-level variables, helping to control for potential confounders in the analysis.

## Discussion

Solitary birth, a phenomenon, where women give birth without any assistance from healthcare professionals or traditional birth attendants, poses serious risks to both maternal and neonatal health. This study disclosed the prevalence, spatial distribution and determinants of solitary birth in Ethiopia using IEDHS 2019.

The prevalence of solitary childbirths in Ethiopia, estimated at 12.73%, with a 95% confidence interval ranging from 11.71% to 13.81%, highlights a significant public health concern. This prevalence indicates that a notable proportion of women are experiencing childbirth alone, which can have profound implications for maternal and infant health outcomes. This statistic also highlights a significant aspect of maternal health in the country, reflecting both the challenges and improvements in childbirth practices.

The results of the spatial autocorrelation analysis revealed significant variation in the distribution of solitary births across Ethiopia, suggesting that these occurrences are not randomly distributed but rather exhibit notable spatial patterns. Socioeconomic factors play a crucial role, as research demonstrates that low-income areas often have limited access to healthcare services, leading to increased rates of solitary births. For instance, a study found that financial constraints and lack of transportation deter low-income women from seeking skilled assistance during childbirth [28]. Additionally, cultural beliefs greatly influence childbirth decisions; many rural communities in Ethiopia prioritize traditional home births, which can result in higher rates of solitary deliveries. A qualitative study highlighted that cultural norms often discourage women from institutional care, contributing to geographic clusters of solitary births [29]. Access to healthcare facilities is another critical factor; the Ethiopian Ministry of Health has reported significant discrepancies in healthcare access between urban and rural areas, with rural regions often lacking adequate maternal healthcare services [30]. This disparity can lead to notable variations in childbirth practices, as women in rural areas may not have the option for institutional delivery, resulting in higher rates of solitary births [30]. Furthermore, the effectiveness of public health interventions varies by region; areas benefiting from targeted maternal health programs, such as community health worker initiatives, often report improved rates of skilled attendance at births. For instance, a study demonstrated that regions with strong community health programs showed lower rates of unassisted births, while areas without such interventions continued to experience high rates of solitary births, thereby contributing to the observed spatial clustering [30].

The findings from the optimized hot spot analysis revealed significant concentrations of solitary births in specific regions of Ethiopia, particularly in the western and southern parts of Oromia, throughout Benishangul-Gumuz, most areas of the SNNPR, and the northwest of Amhara. This clustering suggests that these regions may face systemic challenges that influence maternal health behaviors. Access to healthcare facilities is a critical determinant of maternal health outcomes. Research indicates that rural areas, where many of these hot spots are located, often lack adequate healthcare services, which can lead to higher rates of unassisted births. The Ethiopian Demographic and Health Survey 2016 highlights that limited transportation options and long distances to health facilities can discourage women from seeking assistance during childbirth [31]. Cultural practices and beliefs also significantly impact childbirth decisions. Studies have shown that in many rural communities, traditional norms favor home births and discourage institutional deliveries, leading to higher rates of solitary births [29]. Socioeconomic status is another critical factor; areas with higher poverty rates often exhibit poorer maternal health outcomes. Economic constraints can limit women's access to healthcare services, making solitary births more likely due to financial barriers or the perceived costs associated with institutional deliveries [32]. The effectiveness of public health interventions varies by region. Areas that have benefited from targeted maternal health programs, such as community health worker initiatives, tend to report lower rates of unassisted births. Research indicates that strong community health programs correlate with improved maternal health outcomes, while regions identified as hot spots may lack such interventions, perpetuating higher rates of solitary births [33]. Additionally, educational attainment significantly influences maternal health behaviors, as women with higher education levels are more likely to seek skilled assistance during childbirth. The EDHS data suggest that educational disparities exist, particularly in rural regions, which may contribute to the clustering of solitary births in these hot spot areas [31].

The multilevel logistic regression of this study revealed that having no formal education, not attending ANC visits, and residing in pastoral regions were significantly associated with higher odds of solitary birth in Ethiopia.

Specifically, Women with no formal education exhibited higher odds of experiencing solitary births compared to those with secondary or higher education. This finding aligns with existing literature that emphasizes the role of education in

maternal health outcomes [34–36]. Education empowers women with knowledge about health services, including the importance of skilled attendance during childbirth and the benefits of ANC visits [35]. Studies have shown that educated women are more likely to utilize healthcare services, which can lead to better maternal and neonatal outcomes [34]. In contrast, women without formal education may lack awareness of the risks associated with solitary births and the available healthcare resources.

Additionally, the analysis also revealed that women who did not attend any ANC visits during their pregnancy had significantly higher odds of solitary births. ANC visits are crucial for monitoring the health of both the mother and the fetus, providing essential health education, and preparing for safe delivery. Research indicates that regular ANC attendance is associated with increased likelihood of receiving birth assistance, which is vital for reducing the risks of complications during childbirth [37–39]. The absence of ANC visits may reflect broader issues such as limited access to healthcare services, lack of transportation, or cultural beliefs that discourage seeking medical help during pregnancy [40].

Furthermore, the study found that women living in pastoral regions had a higher likelihood of solitary births compared to those residing in metropolitan areas. Pastoral regions often face unique challenges, including geographical barriers, limited healthcare infrastructure, and cultural practices that may prioritize free births over assisted deliveries [41–43]. Similarly, It has been reported that getting health care services, is difficult for the seasonal migrants who comprise the pastoralist populations [44].These factors contribute to lower utilization of maternal healthcare services in these areas, leading to increased risks associated with solitary births [45]. The disparity in healthcare access between urban and rural or pastoral settings is well-documented, highlighting the need for targeted interventions to improve maternal health services in underserved regions [41–43,45].

## Strength and limmitation of the study

A key strength of this study lies in its methodological rigor, particularly the integration of spatial analysis with multilevel modeling. This combined approach offers a robust framework for identifying geographic disparities and examining both individual- and community-level factors associated with solitary childbirth. Moreover, the focus on solitary childbirth—a largely overlooked yet critical public health issue—fills an important gap in the literature and provides novel insights that can inform targeted maternal health interventions. The identification of hotspot regions further supports the development of geographically tailored strategies, such as community-based education and antenatal care outreach, especially in underserved and high-risk areas.

Despite its strengths, the study is not without limitations. First, the use of data from the 2019 IEDHS, which was collected from a reduced number of enumeration areas compared to the standard DHS, may limit the generalizability of the findings to the entire country. Additionally, the absence of key variables—such as distance to health facilities—in the interim dataset restricts a more comprehensive understanding of access-related barriers to assisted childbirth. The cross-sectional nature of the data further limits the ability to establish causal relationships between the identified determinants and solitary childbirth. Moreover, reliance on self-reported data introduces the possibility of recall bias, as women may underreport or misrepresent their childbirth experiences due to social stigma or limited awareness. Lastly, the analysis may not fully capture regional variations in healthcare practices and access, which could influence the prevalence and determinants of solitary childbirth. These limitations should be considered when interpreting the findings and underscore the need for future research, including longitudinal and qualitative studies, to deepen understanding of this issue.

## Implication of the study

**Implications for maternal health.** The relatively high prevalence of solitary childbirths suggests that many women may lack adequate support during labor and delivery. This absence of support can lead to increased stress and anxiety, potentially complicating the childbirth experience. Support from family members, healthcare providers, or

birthing companions is crucial in promoting positive birth experiences. Additionally, the data may reflect broader issues regarding access to healthcare services. Women giving birth alone might indicate barriers such as geographical distance from healthcare facilities, financial constraints, or lack of transportation. Understanding these barriers is essential for developing policies aimed at improving maternal healthcare access. Solitary childbirth may also be influenced by cultural beliefs and practices. In some communities, women may prefer or be expected to give birth alone due to traditional norms. Addressing these cultural dimensions through community engagement and education could promote safer childbirth practices.

**Implications for infant health.**  The risks associated with solitary childbirth can extend to the newborn. Without professional assistance, there may be delays in addressing complications that could arise during delivery, potentially affecting neonatal outcomes. Ensuring that women have access to skilled birth attendants is crucial for reducing infant morbidity and mortality. Furthermore, women who give birth alone may also face challenges in accessing postnatal care for themselves and their infants. Ensuring follow-up visits and support services can be vital in promoting maternal and child health after delivery.

**Recommendations for policy and practice.**  There is a need for policies that enhance access to maternal healthcare services, particularly in underserved areas. This includes increasing the number of skilled birth attendants and improving transportation options for expectant mothers. Initiatives aimed at educating communities about the importance of skilled assistance during childbirth can help shift cultural perceptions and encourage more women to seek help. Continued research is essential to monitor trends in solitary childbirth and to understand the underlying factors contributing to this prevalence. This data can inform targeted interventions and policies.

## Conclusion

A notable proportion of women are experiencing childbirth alone, which highlights a significant aspect of maternal health in the country, reflecting both the challenges and improvements in childbirth practices. The distribution of solitary births exhibited spatial clustering with its hotspot areas located in western and southern parts of Oromia, all of Benishangul-Gumuz, most parts of the SNNPR, and west of Amhara regions. Lack of education, not having an ANC visit, and being a resident of pastoral regions were significant determinants of solitary birth. The implementation of maternal and child health strategies in Ethiopia could benefit from considering the hotspot areas and determinants of solitary birth.

## Supporting information

**S1 Data.  Mini data.**
(ZIP)

## Author contributions

**Conceptualization:** Tadesse Tarik Tamir, Berhan Tekeba, Alebachew Ferede Zegeye, Deresse Abebe Gebrehana, Mulugeta Wassie, Gebreeyesus Abera Zeleke.

**Data curation:** Tadesse Tarik Tamir, Berhan Tekeba, Alebachew Ferede Zegeye, Deresse Abebe Gebrehana, Mulugeta Wassie, Gebreeyesus Abera Zeleke, Enyew Getaneh Mekonen.

**Formal analysis:** Tadesse Tarik Tamir, Alebachew Ferede Zegeye, Mulugeta Wassie.

**Investigation:** Tadesse Tarik Tamir, Berhan Tekeba, Alebachew Ferede Zegeye, Deresse Abebe Gebrehana, Mulugeta Wassie, Gebreeyesus Abera Zeleke, Enyew Getaneh Mekonen.

**Methodology:** Tadesse Tarik Tamir, Berhan Tekeba, Alebachew Ferede Zegeye, Deresse Abebe Gebrehana, Gebreeyesus Abera Zeleke, Enyew Getaneh Mekonen.

**Resources:** Tadesse Tarik Tamir.

**Software:** Tadesse Tarik Tamir, Berhan Tekeba, Enyew Getaneh Mekonen.

**Validation:** Tadesse Tarik Tamir, Berhan Tekeba, Alebachew Ferede Zegeye, Deresse Abebe Gebrehana, Mulugeta Wassie, Enyew Getaneh Mekonen.

**Visualization:** Tadesse Tarik Tamir, Berhan Tekeba, Alebachew Ferede Zegeye, Deresse Abebe Gebrehana, Mulugeta Wassie, Gebreeyesus Abera Zeleke, Enyew Getaneh Mekonen.

**Writing – original draft:** Tadesse Tarik Tamir.

**Writing – review & editing:** Tadesse Tarik Tamir, Berhan Tekeba, Alebachew Ferede Zegeye, Deresse Abebe Gebrehana, Mulugeta Wassie, Gebreeyesus Abera Zeleke, Enyew Getaneh Mekonen.

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
