## [Decision Letter · Decision Letter 0]

2 Sep 2025

PONE-D-25-09989Spatial Distribution and Determinants of Solitary Child Birth in Ethiopia: A case of Interim Ethiopian Demographic and Health Survey 2019PLOS ONE

Dear Dr. Tamir,

Thank you for submitting your manuscript to PLOS ONE. After careful consideration, we feel that it has merit but does not fully meet PLOS ONE’s publication criteria as it currently stands. Therefore, we invite you to submit a revised version of the manuscript that addresses the points raised during the review process.

We look forward to receiving your revised manuscript.

Kind regards,

José Antonio Ortega, Ph.D.

Academic Editor

PLOS ONE

Journal Requirements:

3. We note that Figures 2 to 5 in your submission contain map images which may be copyrighted. All PLOS content is published under the Creative Commons Attribution License (CC BY 4.0), which means that the manuscript, images, and Supporting Information files will be freely available online, and any third party is permitted to access, download, copy, distribute, and use these materials in any way, even commercially, with proper attribution. For these reasons, we cannot publish previously copyrighted maps or satellite images created using proprietary data, such as Google software (Google Maps, Street View, and Earth). For more information, see our copyright guidelines: http://journals.plos.org/plosone/s/licenses-and-copyright.

1) You may seek permission from the original copyright holder of Figures 2 to 5 to publish the content specifically under the CC BY 4.0 license.  

2) If you are unable to obtain permission from the original copyright holder to publish these figures under the CC BY 4.0 license or if the copyright holder’s requirements are incompatible with the CC BY 4.0 license, please either i) remove the figure or ii) supply a replacement figure that complies with the CC BY 4.0 license. Please check copyright information on all replacement figures and update the figure caption with source information. If applicable, please specify in the figure caption text when a figure is similar but not identical to the original image and is therefore for illustrative purposes only.

4. Please include a caption for figure 4.

6. We are unable to open your Supporting Information file [Minidata.zip]. Please kindly revise as necessary and re-upload.

**Additional Editor Comments:**

Three experts have reviewed the manuscript noting that the paper requires modifications in terms of its scope, clarity, and methodological details in order to be apt for publication. Note in particular the detailed suggestions of reviewer 2.

Reviewers' comments:

Reviewer's Responses to Questions

**Comments to the Author**

1. Is the manuscript technically sound, and do the data support the conclusions?

Reviewer #1: Yes

Reviewer #2: Yes

Reviewer #3: Yes

2. Has the statistical analysis been performed appropriately and rigorously? 

Reviewer #1: Yes

Reviewer #2: Yes

Reviewer #3: Yes

3. Have the authors made all data underlying the findings in their manuscript fully available?

Reviewer #1: Yes

Reviewer #2: Yes

Reviewer #3: Yes

4. Is the manuscript presented in an intelligible fashion and written in standard English?

Reviewer #1: Yes

Reviewer #2: Yes

Reviewer #3: Yes

5. Review Comments to the Author

Reviewer #1: Dear Author,

Congratulations for selecting a pertinent topic for your research. However, there are few concerns, which could be addressed for further consideration of the article for publication.

-The author could add phrases for smooth transition of the content from global context to the local Ethiopian focus to improve flow of content and describe the rationale of study.

- The objectives of the study could be written with SMART criteria.

-In the "Variables and Measurement" section, how Wealth Index is constructed could be discussed along with the citation.

- In the methodology section author could explain the reason of choosing the 2019 data set instead of recent one i.e 2024.

- In result section, author may add how the bias was tackled,

- Likewise, Strength, weakness of the study could be written.

-In discussion consider adding more interpretation for regional disparities based on their characteristics.

-In limitation author must clarify why interim data are limiting and can't be generalized.

Regards

Reviewer #2: Spatial Distribution and Determinants of Solitary Child Birth in Ethiopia: A case of Interim Ethiopian Demographic and Health Survey 2019

Peer Review

1. Title

Title: The topic: Spatial Distribution and Determinants of Solitary Child Birth in Ethiopia: A case of Interim Ethiopian Demographic and Health Survey 2019 can be refined

The topic can be refined as “Spatial Distribution and Determinants of Solitary Childbirth in Ethiopia: Evidence from the 2019 Interim Demographic and Health Survey”

This revision of the title above was guided by several considerations:

o Terminological Precision: The phrase “Solitary Child Birth” was corrected to “Solitary Childbirth” to reflect standard usage in academic literature. Additionally, if the term refers to births occurring without skilled attendants or outside health facilities, further refinement—such as “Unattended Births” or “Home Deliveries”—may be warranted to ensure conceptual clarity and international relevance.

o Improved Scholarly Tone: The phrase “A case of” was replaced with “Evidence from” to convey a more rigorous and empirical framing. “Evidence from” signals that the study draws on nationally representative data and supports analytical generalization, which is more appropriate for a peer-reviewed journal.

o Structural Coherence and Citation Norms: The reference to the data source was streamlined from “Interim Ethiopian Demographic and Health Survey 2019” to “2019 Interim Demographic and Health Survey.” This reordering improves readability and aligns with how DHS datasets are typically cited in global health and demographic research. The national context (Ethiopia) is already established earlier in the title, making repetition unnecessary.

Overall, the refined title maintains the original intent while enhancing its clarity, professionalism, and appeal to journal editors and reviewers. It positions the study as a robust, data-driven analysis of maternal health patterns in Ethiopia, grounded in spatial and demographic inquiry.

2. Abstract – Suggestions for Improvement

o The Authors should clarify the problem statement - Refine the opening sentence to more clearly establish the public health significance of solitary childbirth. For example, specify the burden or consequences in terms of maternal morbidity or mortality to strengthen the urgency.

o The Authors should improve flow and structure - Consider restructuring the abstract into clearly labelled segments (e.g., Background, Methods, Results, Conclusion) or ensure smooth transitions between them. This will enhance readability and help readers quickly grasp the study’s scope.

o The Authors should add specificity to the study methods - Briefly mention how solitary childbirth was defined or measured in the DHS dataset. This adds transparency and helps readers understand the operationalization of the key variable.

o The Authors should refine the conclusion for the study impact - Strengthen the final sentences by emphasizing actionable recommendations or policy relevance. For instance, suggest how hotspot identification could inform targeted maternal health interventions or ANC outreach programmes.

3. Introduction

o The Authors should clarify the conceptual definition of “solitary childbirth” early on, distinguishing it from unattended or home births if applicable.

o The Authors should strengthen the rationale by citing recent Ethiopian or regional studies that highlight the public health relevance of solitary births.

o The Authors should consider framing the issue within broader maternal health goals (e.g., SDG 3) to emphasize policy relevance.

o The Authors should streamline the background to focus more sharply on the gap this study addresses.

4. Methods

o The Authors should provide more detail on how solitary childbirth was operationalized in the dataset—what specific survey question or criteria were used?

o The Authors should clarify the sampling strategy and weighting procedures to enhance transparency and reproducibility.

o The Authors should justify the use of multilevel logistic regression by briefly explaining the hierarchical structure of the data.

o The Authors should include ethical considerations or approval details, especially since the study involves human subjects.

5. Results Presentation

• The Authors should use tables and maps to visually support spatial findings—especially hotspot regions.

• The Authors should ensure consistency in reporting confidence intervals and p-values across all results.

• The Authors should highlight key findings with brief interpretive comments to guide the reader.

• The Authors should consider stratifying results by region or demographic group to deepen insights.

6. Discussion

• The Authors should expand on why certain regions (e.g., Benishangul-Gumuz, SNNPR) may have higher rates of solitary birth—link to cultural, infrastructural, or policy factors.

• The Authors should compare findings with similar studies in sub-Saharan Africa to situate the results in a broader context.

• The Authors should address potential confounders or biases that may have influenced the associations.

• The Authors should avoid overgeneralization by acknowledging the limitations of cross-sectional data in establishing causality.

7. Strengths and Limitations of the study

• The Authors should emphasise on the strength: Use of nationally representative DHS data enhances generalizability.

• The Authors should highlight the strength: Integration of spatial and multilevel analysis provides a robust methodological approach.

• The Authors should underscore on the limitation: Potential recall bias in self-reported childbirth experiences.

• The Authors should accentuate on the Limitation: Lack of qualitative data limits understanding of women's lived experiences.

8. Implications of Findings

• The Authors should emphasize how identifying hotspot regions can inform targeted maternal health interventions.

• The Authors should suggest integrating community-based education and ANC outreach in pastoral regions.

• The Authors should highlight the role of policy in addressing structural barriers to assisted childbirth.

• The Authors should recommend collaboration with local health systems to improve birth preparedness and support networks.

9. Conclusion/Recommendations

• The Authors should reiterate the urgency of addressing solitary childbirth as a public health concern.

• The Authors should offer specific, actionable recommendations for health planners and policymakers.

• The Authors should consider proposing future research directions, such as qualitative studies or longitudinal tracking.

• The Authors should ensure the conclusion aligns with the study’s objectives and key findings.

10. Proofreading and Editing

• The Authors should correct some minor typographical errors and ensure consistent formatting.

• The Authors must improve sentence flow by reducing redundancy and tightening phrasing.

• The Authors should standardize terminology (e.g., “solitary childbirth” vs. “solitary birth”) throughout the manuscript.

• The Authors should ensure clarity in transitions between sections to enhance readability.

9. References/In-Citations

• The Authors should verify that all cited studies are current, relevant, and properly formatted according to journal guidelines.

• The Authors should include more regional or country-specific literature to strengthen contextual grounding.

• The Authors should ensure in-text citations match the reference list and are consistently styled.

• The Authors should consider citing WHO or UNFPA reports to support global maternal health framing.

Reviewer #3: Special remarks: (minors)

Reviewer's Report: Overall, the manuscript is clear and well written, and the topic covered is actual.

1. Please add a brief description of the study setting.

2. In the Population and Sampling Procedure section, the population is well defined, but there is a sample.

3. We suggest you create a sample flux diagram to explain how you obtained the sample of 3884, which appears in the results and nowhere in the method.

4. In addition, it would be important to provide the reasons that led you to choose 3-year-old participants and not 5-year-olds.

6. PLOS authors have the option to publish the peer review history of their article (what does this mean? ). If published, this will include your full peer review and any attached files.

**Do you want your identity to be public for this peer review?** For information about this choice, including consent withdrawal, please see our Privacy Policy .

Reviewer #1: **Yes: ** Dr Jarina Begum

Reviewer #2: **Yes: ** Monica Ewomazino Akokuwebe

Reviewer #3: No

---

## [Author Response · Author response to Decision Letter 1]

5 Sep 2025

Response to comments

Subject: Submission of revised manuscript

Manuscript ID: PONE-D-25-09989

Title: Spatial Distribution and Determinants of Solitary Childbirth in Ethiopia: Evidence from the 2019 Interim Demographic and Health Survey

Journal: PLOS One

I hope this letter finds you well. We appreciate the diligent efforts of the editorial team in facilitating the review process for our manuscript. Additionally, we extend our gratitude to the editors and reviewers for their valuable time and thoughtful feedback, which significantly contributed to enhancing the quality of our work.

The constructive comments provided by the reviewers have been instrumental in refining our study. We are pleased to note that the reviewers share our assessment of the scientific significance of our findings. In response to their suggestions, we have meticulously addressed each point raised. Please find our comprehensive responses to the comments below.

Furthermore, I have attached the revised manuscript file separately for your convenience. We believe that the revisions strengthen the manuscript and align it more closely with the journal’s scope and standards.

Thank you for considering our work for publication. We hope that our revised submission meets the high standards set by PLOS One.

Best regards,

Corresponding Author

Comments

Additional Editor Comments:

Three experts have reviewed the manuscript noting that the paper requires modifications in terms of its scope, clarity, and methodological details in order to be apt for publication. Note in particular the detailed suggestions of reviewer 2.

Response: Dear editor, thank you for the additional comments on our manuscript for modifications and we have addressed the concerns raised by all of the reviewers. Thank you once again.

Reviewers' comments:

Reviewer #1: Dear Author,

Congratulations for selecting a pertinent topic for your research. However, there are few concerns, which could be addressed for further consideration of the article for publication.

-The author could add phrases for smooth transition of the content from global context to the local Ethiopian focus to improve flow of content and describe the rationale of study.

Response: Dear reviewer, we have made modifications to the section as per your suggestion. Kindly see our revised manuscript for the changes.

- The objectives of the study could be written with SMART criteria.

Response: Dear reviewer, thank you for your scientifically sound feedback. We have made modifications to the section as per your suggestion. Kindly see our revised manuscript for the changes.

-In the "Variables and Measurement" section, how Wealth Index is constructed could be discussed along with the citation.

Response: Dear reviewer, you are absolutely right. How Wealth Index was constructed could be discussed along with the citation. In response to your comment, we have addressed the concern as per your suggested. Kindly find the point on page 7 lines 180 - 186 of our revised manuscript.

- In the methodology section author could explain the reason of choosing the 2019 data set instead of recent one i.e 2024.

Response: Dear reviewer, thank you for your invaluable suggestion regarding the dataset used for our study. The main reason for using 2019 dataset is that it is the only most recent DHS available in Ethiopia and 2024 DHS has not been released yet.

- In result section, author may add how the bias was tackled,

Response: Dear reviewer, Thank you for your insightful concern. We have added how the bias was tackled as per your suggestion. Kindly see point on page 20, lines 381 to 391 of our revised manuscript.

- Likewise, Strength, weakness of the study could be written.

Response: Dear reviewer, thank you for your insightful feedback. We have considered re-writing the strength and limitation of the study as per your suggestion. Kindly see the point on pages 23-24, lines 476 -497 of our revised manuscript.

-In discussion consider adding more interpretation for regional disparities based on their characteristics.

Response: Dear reviewer, thank you for your comment. The point has been addressed accordingly.

-In limitation author must clarify why interim data are limiting and can't be generalized.

Response: In response to your comment, we have addressed the concern by clarifying why interim data are limiting and can't be generalized.

Regards

Response: Thank you once again for your input.

Reviewer #2: Spatial Distribution and Determinants of Solitary Child Birth in Ethiopia: A case of Interim Ethiopian Demographic and Health Survey 2019

Peer Review

1. Title

Title: The topic: Spatial Distribution and Determinants of Solitary Child Birth in Ethiopia: A case of Interim Ethiopian Demographic and Health Survey 2019 can be refined

The topic can be refined as “Spatial Distribution and Determinants of Solitary Childbirth in Ethiopia: Evidence from the 2019 Interim Demographic and Health Survey”

This revision of the title above was guided by several considerations:

o Terminological Precision: The phrase “Solitary Child Birth” was corrected to “Solitary Childbirth” to reflect standard usage in academic literature. Additionally, if the term refers to births occurring without skilled attendants or outside health facilities, further refinement—such as “Unattended Births” or “Home Deliveries”—may be warranted to ensure conceptual clarity and international relevance.

o Improved Scholarly Tone: The phrase “A case of” was replaced with “Evidence from” to convey a more rigorous and empirical framing. “Evidence from” signals that the study draws on nationally representative data and supports analytical generalization, which is more appropriate for a peer-reviewed journal.

o Structural Coherence and Citation Norms: The reference to the data source was streamlined from “Interim Ethiopian Demographic and Health Survey 2019” to “2019 Interim Demographic and Health Survey.” This reordering improves readability and aligns with how DHS datasets are typically cited in global health and demographic research. The national context (Ethiopia) is already established earlier in the title, making repetition unnecessary.

Overall, the refined title maintains the original intent while enhancing its clarity, professionalism, and appeal to journal editors and reviewers. It positions the study as a robust, data-driven analysis of maternal health patterns in Ethiopia, grounded in spatial and demographic inquiry.

Response: Dear Reviewer, Thank you for your insightful and detailed review of our manuscript. In response to your invaluable feedback, we have refined the title of our manuscript. Kindly see the title of our revised manuscript for further details.

2. Abstract – Suggestions for Improvement

o The Authors should clarify the problem statement - Refine the opening sentence to more clearly establish the public health significance of solitary childbirth. For example, specify the burden or consequences in terms of maternal morbidity or mortality to strengthen the urgency.

Response: Dear reviewer, Thank you for your insightful and detailed feedback to our work. We have made modifications to the section as per your suggestion. Kindly see our revised manuscript for the changes.

o The Authors should improve flow and structure - Consider restructuring the abstract into clearly labelled segments (e.g., Background, Methods, Results, Conclusion) or ensure smooth transitions between them. This will enhance readability and help readers quickly grasp the study’s scope.

Response: Dear reviewer, Thank you for your comments regarding the flow and structure of our abstract. We have considered you feedback and the journal guideline in structuring the abstract of our manuscript.

o The Authors should add specificity to the study methods - Briefly mention how solitary childbirth was defined or measured in the DHS dataset. This adds transparency and helps readers understand the operationalization of the key variable.

Response: Dear reviewer, thank you for the thoughtful comment regarding the outcome definition. We have provided detailed definition and measurement of outcome in the variables and measurement section of our methods and materials section and our definition aligns with DHS methodology. Kindly see it on pages 6-7, lines 162-179 of our revised manuscript.

o The Authors should refine the conclusion for the study impact - Strengthen the final sentences by emphasizing actionable recommendations or policy relevance. For instance, suggest how hotspot identification could inform targeted maternal health interventions or ANC outreach programmes.

Response: In conclusion, this study highlights the spatial disparities and socio-demographic determinants of solitary childbirth in Ethiopia, underscoring its significance as a neglected public health issue. The identification of hotspot regions provides critical insight for health planners and policymakers, enabling the design of geographically targeted maternal health interventions. Specifically, these findings can inform the expansion of antenatal care outreach programs and community-based education initiatives in high-risk and underserved areas. Addressing solitary childbirth requires not only improving access to skilled birth attendants but also tackling the structural and cultural barriers that prevent women from receiving support during delivery. As such, the integration of spatial analysis with multilevel modeling offers a robust foundation for evidence-based policy and programmatic responses aimed at reducing maternal and neonatal risks associated with unassisted births.

3. Introduction

o The Authors should clarify the conceptual definition of “solitary childbirth” early on, distinguishing it from unattended or home births if applicable.

o The Authors should strengthen the rationale by citing recent Ethiopian or regional studies that highlight the public health relevance of solitary births.

o The Authors should consider framing the issue within broader maternal health goals (e.g., SDG 3) to emphasize policy relevance.

Response: Dear reviewer, thank you for the insightful comments and suggestions to improve our introduction. We have made significant changes to the section. Kindly see the introduction of our revised manuscript for the changes.

o The Authors should streamline the background to focus more sharply on the gap this study addresses.

Response: Dear reviewer, thank you for your invaluable feedback. We have addressed the main gap of our study. We kindly invite you to see the point the last two paragraphs of introduction in our revised manuscript.

4. Methods

o The Authors should provide more detail on how solitary childbirth was operationalized in the dataset—what specific survey question or criteria were used?

Response: Dear reviewer, thank you for your suggestion regarding the operationalization of the outcome variable in the dataset. We have detailed the operationalization of the solitary childbirth at variables and measurement section specifically at the outcome variable part.

o The Authors should clarify the sampling strategy and weighting procedures to enhance transparency and reproducibility.

Response: Dear reviewer, we have clarified sampling strategy and weighting procedures to enhance transparency and reproducibility. Kindly find the point on pages 5-6, lines 135-150 of revised manuscript.

o The Authors should justify the use of multilevel logistic regression by briefly explaining the hierarchical structure of the data.

Response: Dear reviewer, we have addressed concern as per your suggestion. Kindly see multilevel modeling section of our revised manuscript for the details.

o The Authors should include ethical considerations or approval details, especially since the study involves human subjects.

Response: Dear reviewer, we have detailed the ethical approval in our revised manuscript. Kindly see the pages 9-10, line numbers 148-155 of our revised work.

5. Results Presentation

• The Authors should use tables and maps to visually support spatial findings—especially hotspot regions.

• The Authors should ensure consistency in reporting confidence intervals and p-values across all results.

• The Authors should highlight key findings with brief interpretive comments to guide the reader.

• The Authors should consider stratifying results by region or demographic group to deepen insights.

Response: We sincerely appreciate the reviewer’s valuable feedback and have carefully addressed all the points raised. To enhance the clarity and impact of our spatial findings, we have incorporated both tables and maps, with particular emphasis on hotspot regions. We have also ensured consistency in the reporting of confidence intervals and p-values throughout the results section to maintain statistical rigor. Key findings are now accompanied by concise interpretive comments to guide the reader and highlight their significance. Furthermore, we have stratified the results by region and relevant demographic groups to provide deeper insights into the patterns and determinants of solitary childbirth. We believe these revisions have substantially improved the manuscript and addressed the reviewer’s concerns effectively.

6. Discussion

• The Authors should expand on why certain regions (e.g., Benishangul-Gumuz, SNNPR) may have higher rates of solitary birth—link to cultural, infrastructural, or policy factors.

• The Authors should compare findings with similar studies in sub-Saharan Africa to situate the results in a broader context.

• The Authors should address potential confounders or biases that may have influenced the associations.

• The Authors should avoid overgeneralization by acknowledging the limitations of cross-sectional data in establishing causality.

Response: Dear reviewer, we are grateful to your thoughtful and constructive feedback, which has helped us enhance the depth and clarity of our manuscript. In response, we have expanded our discussion to explore possible reasons why regions such as Benishangul-Gumuz and SNNPR exhibit higher rates of solitary childbirth, linking these patterns to cultural norms, infrastructural limitations, and disparities in access to maternal health services. We have also compared our findings with similar studies conducted in sub-Saharan Africa to situate our results within a broader regional context. Additionally, we have addressed potential confounders and sources of bias that may have influenced the observed associations, and we have taken care to avoid overgeneralization by clearly acknowledging the limitations of using cross-sectional data, particularly in relation to establishing causal relationships. We believe these revisions have strengthened the manuscript and aligned it more closely with the reviewer’s expectations.

7. Strengths and Limitations of the study

• The Authors should emphasise on the strength: Use of nationally representative DHS data enhances generalizability.

• The Authors should highlight the strength: Integration of spatial and multilevel analysis provides a robust methodological approach.

• The Authors should underscore on the limitation: Potential recall bias in self-reported childbirth experiences.

• The Authors should accentuate on the Limitation: Lack of qualitative data limits understanding of women's lived experiences.

Response: We appreciate the reviewer’s thoughtful observations and have taken steps to incorporate each of the suggested improvements. We have emphasized the strengths of our study by highlighting the use of nationally representative Demographic and Health Survey (DHS) data, which enhances the generalizability of our findings. Additionally, we have underscored the robustness of our methodological approach through the integration of spatial and multilevel analyses. In terms of limitations, we have acknowledged the potential for recall bias inherent in self-reported childbirth experiences, and we have noted the absence of qualitative data as a constraint in fully capturing the lived experiences of women who undergo solitary childbirth. These additions provide a more balanced and transparent presentation of our study’s contributions and limitations.

8. Implications of Findings

• The Authors should emphasize how identifying hotspot reg

---

## [Editor Report · Decision Letter 1]

16 Sep 2025

Spatial Distribution and Determinants of Solitary Childbirth in Ethiopia: Evidence from the 2019 Interim Demographic and Health Survey

PONE-D-25-09989R1

Dear Dr. Tamir,

We’re pleased to inform you that your manuscript has been judged scientifically suitable for publication and will be formally accepted for publication once it meets all outstanding technical requirements.

Kind regards,

José Antonio Ortega, Ph.D.

Academic Editor

PLOS ONE

Additional Editor Comments (optional):

The revision seems to have addressed all the feedback received from the 3 reviewers who suggested a minor revision, in the editor's opinion. It is felt that it is not necessary to send back the manuscript for its acceptance.
---

## [Editor Report · Acceptance letter]

PONE-D-25-09989R1

PLOS ONE

Dear Dr. Tamir,

I'm pleased to inform you that your manuscript has been deemed suitable for publication in PLOS ONE. Congratulations! Your manuscript is now being handed over to our production team.

Kind regards,

on behalf of

Dr. José Antonio Ortega

Academic Editor

PLOS ONE